# HICO-DET-SG and V-COCO-SG: New Data Splits for Evaluating the Systematic Generalization Performance of Human-Object Interaction Detection Models

## Abstract

Human-Object Interaction (HOI) detection is a task to localize humans and objects in an image and predict the interactions in human-object pairs. In real-world scenarios, HOI detection models are required systematic generalization, i.e., generalization to novel combinations of objects and interactions, because the train data are expected to cover a limited portion of all possible combinations. However, to our knowledge, no open benchmarks or previous work exist for evaluating the systematic generalization performance of HOI detection models. To address this issue, we created two new sets of HOI detection data splits named HICO-DET-SG and V-COCO-SG based on the HICO-DET and V-COCO datasets, respectively. When evaluated on the new data splits, the representative HOI detection models performed much more poorly than when evaluated on the original splits. This reveals that systematic generalization is a challenging goal in HOI detection. By analyzing the evaluation results, we also gain insights for improving the systematic generalization performance and identify four possible future research directions. We hope that our new data splits and presented analysis will encourage further research on systematic generalization in HOI detection.

## 1 Introduction

Human-Object Interaction (HOI) detection is a task to localize humans and objects in an image and predict the interactions in human-object pairs. HOI detection has been attracting large interest in computer vision, as it is useful for various applications such as self-driving cars, anomaly detection, analysis of surveillance video, and so on. The outputs of this task are typically represented as <human, **interaction**, object> triplets. The publication of HOI detection datasets (Koppula et al., 2013; Everingham et al., 2014; Gupta & Malik, 2015; Chao et al., 2018; Gu et al., 2018; Chiou et al., 2021) has triggered a large number of studies on this task (Gao et al., 2018; Gkioxari et al., 2018; Li et al., 2019; Gao et al., 2020; Li et al., 2020; Liao et al., 2020; Kim et al., 2021; Chen & Yanai, 2021; Tamura et al., 2021; Zhang et al., 2021; Chen et al., 2021; Zou et al., 2021; Zhang et al., 2022; Liao et al., 2022; Ma et al., 2023).

HOI detection is an advanced computer vision task as it requires a model not only to localize humans and objects but also to predict interactions between them. Moreover, humans can have different interactions with the same object (e.g., **wash** a horse and **walk** a horse) and the same interaction with different objects (e.g., wash a horse and wash a car). In real-world scenarios, the train data will likely cover a limited portion of all possible combinations of objects and interactions of interest. Thus, HOI detection models must be generalizable to novel combinations of known objects and interactions.

Such generalization to novel combinations of known concepts, called systematic generalization, is a highly desired property for machine learning models. The systematic generalization performance is evaluated in various tasks such as sequence-to-sequence parsing (Lake & Baroni, 2018), language understanding (Ruis et al., 2020; Bergen et al., 2021), visual properties extraction (Ullman et al., 2021), and visual question answering (Johnson et al., 2017; Bahdanau et al., 2019; 2020; D'Amario et al., 2021; Hsu et al., 2022; Yamada

Figure 1: Illustration of a data split for evaluating the systematic generalization performance of Human-Object Interaction (HOI) detection models. All images and annotations are selected from HICO-DET-SG split3. The train data consists of combinations such as <human, **wash**, car>, <human, **wash**, elephant>, <human, **walk**, horse>, and <human, **straddle**, horse>. After trained on such data, an HOI detection model is tested whether it can generalize to novel combinations in the test data such as <human, **wash**, horse>. To systematically generalize to such novel combinations, the model must learn the visual cues of the object (in this case, horse) and the interaction (in this case, **wash**) independently of the specifically paired interaction/object classes in the train data.

et al., 2023; Kamata et al., 2023). However, to our knowledge, no open benchmarks or evaluation studies have been published for the systematic generalization in HOI detection.

The existing HOI detection datasets cannot be used for evaluating the systematic generalization performance for novel combinations of known objects and interactions as they are, because their train and test data provide the same object-interaction combinations. On such datasets, a model might predict an interaction class based solely on the paired object class or an object class based solely on the paired interaction class rather than by capturing the visual cues such as human posture and positional relations.

In this paper, we introduce two new sets of HOI detection data splits named HICO-DET-SG and V-COCO-SG, which we created based on the HICO-DET (Chao et al., 2018) and V-COCO (Gupta & Malik, 2015) datasets, respectively, for evaluating the systematic generalization (SG) capabilities of HOI detection models. An illustration of such data split is shown in Figure 1.

To ensure that the test performance is not an artifact of a specific selection of combinations in the train and test data, we prepared three distinct train-test splits of HICO-DET-SG and V-COCO-SG, respectively. We evaluated recent representative HOI detection models on our data splits and found a large degradation from the performances on the original data splits. We also analyzed the results and gained insights to improve systematic generalization performance of HOI detection models.

Our contributions are summarized below:

- We created two new sets of HOI detection data splits with no overlapping object-interaction combinations in train and test data, which serve for studying systematic generalization in HOI detection.

- We evaluated the systematic generalization performances of representative HOI detection models on our new data splits and found large decreases in the test performance from those on the original splits; this reveals that the systematic generalization is a challenging goal in HOI detection.

- We derived four possible future directions to improve systematic generalization performance in HOI detection based on the analysis and considerations on our experimental results and related work: 1) increasing the diversity of the train data, 2) introducing two-stage or other modular structures into a model, 3) utilizing pretraining, and 4) integrating commonsense knowledge from external natural language resources.

The JSON files determining HICO-DET-SG and V-COCO-SG and the source code that created the files are publicly available at a GitHub repository. For the review, they can be assessed via the following anonymized repository `https://anonymous.4open.science/r/hoi_sg-58CE/`.

## 2 Related work

In this section, we first briefly review the HOI detection task. Then we explain the related work to the new research topic we deal with in this study, namely, systematic generalization in HOI detection.

### 2.1 Overview of Human-Object Interaction (HOI) detection

As explained in the Introduction, HOI detection is a task to localize humans and objects in an image and predict the interactions between them. The HICO-DET (Chao et al., 2018) and V-COCO (Gupta & Malik, 2015) are the two most popular datasets for HOI detection. Preceding to the HICO-DET dataset, the HICO dataset (Chao et al., 2015) was created for HOI recognition task, which classifies an object and a human's interaction with that object in an image without bounding-boxes. Subsequently, HICO-DET dataset was created based on HICO by adding bounding-boxes around the humans and objects in the images. Moreover, one image in the dataset is associated with multiple human, object, and interaction annotations. The V-COCO dataset was created based on the Microsoft COCO object detection dataset (Lin et al., 2014) by adding annotations of interactions (verbs). Statistics of the HICO-DET and V-COCO datasets are given in Table 1.

By definition, HOI detection consists of two tasks: localizing the human and object instances in a given image and predicting the interactions between them. Accordingly, there are two types of model architectures to solve the HOI detection task: two-stage model and one-stage model. Two-stage models (Gao et al., 2018; Li et al., 2019; Gao et al., 2020; Li et al., 2020; Liao et al., 2022; Zhang et al., 2022) detect humans and objects at the first stage and then classify the interactions in all human-object pairs at the second stage. Aiming to improve both the instance detection and interaction classification via multi-task learning and to reduce inference time, one-stage models (Gkioxari et al., 2018; Liao et al., 2020) have been proposed recently and gained popularity over two-stage models. Some recent one-stage models (Kim et al., 2021; Tamura et al., 2021; Chen & Yanai, 2021; Zhang et al., 2021; Chen et al., 2021; Zou et al., 2021; Ma et al., 2023) are based on the Transformer architecture (Vaswani et al., 2017), which is designed to capture long-range relationships in an input and has been successful in natural language processing (Kalyan et al., 2021; Lin et al., 2022) and computer vision (Khan et al., 2021).

HOI detection is closely related to Scene Graph Generation (SGG) (Johnson et al., 2015), which is a task to generate a visually-grounded scene graph that most accurately correlates with an image. A scene graph consists of nodes corresponding to object bounding boxes with their object categories, and edges indicating the pairwise relations between objects. While HOI detection is closely related to SGG, HOI detection differs from SGG in two main ways. First, the subjects in SGG can be of any type (humans, cars, etc.), whereas in HOI detection they are only humans. Second, the relations in SGG can be both positional relations (e.g., next to) and actions (e.g., play with), whereas in HOI detection they only consist of the latter. Therefore, HOI detection can be regarded as a subset of SGG in a sense. On the other hand, HOI detection can also be regarded as a task focusing on measuring a model's ability for complex scene understanding, because action recognition requires additional information, not merely the locations of humans and objects.

## 2.2 Studies related to systematic generalization in HOI detection

Systematic generalization (Lake & Baroni, 2018; Bahdanau et al., 2019; Ruis et al., 2020; Bahdanau et al., 2020; Bergen et al., 2021; D'Amario et al., 2021; Yamada et al., 2023; Kamata et al., 2023), also referred to as compositional generalization (Johnson et al., 2017; Kim & Linzen, 2020; Hsu et al., 2022) or combinatorial generalization (Vankov & Bowers, 2020; Ullman et al., 2021), is a special case of Out-of-Distribution (OoD) generalization, i.e., generalization to data distributions that differ from the train data (Teney et al., 2020; Hendrycks et al., 2021; Shen et al., 2021; Ye et al., 2022). Among many types of OoD generalization, systematic generalization has been particularly considered as a hallmark of human intelligence and contrasted to the properties of artificial neural networks in each era (Fodor & Pylyshyn, 1988; Marcus, 2001; van der Velde et al., 2004; Lake et al., 2017; 2019; Baroni, 2020; Smolensky et al., 2022). Recently, the systematic generalization capability of deep neural networks, including Transformer-based models, has been actively studied in various tasks, as explained in the Introduction.

In HOI detection, HICO-DET dataset provides rare-triplets evaluation to measure the few-shot generalization ability of the models (rare-triplets are defined as those which appear less than 10 times in the train data). Generalization to rare-triplets is a type of OoD generalization and some existing work (Baldassarre et al., 2020; Ji et al., 2021) attempted to improve this performance. However, to our knowledge, no benchmarks or previous work have tackled systematic generalization, i.e., zero-shot generalization, in HOI detection. We present the first data splits for evaluating the systematic generalization performance and benchmark results of the representative HOI models in this paper.

In SGG, there are some previous work to evaluate (Tang et al., 2020) and improve (Lu et al., 2016; Kan et al., 2021) systematic generalization for new combinations of subject, relation and object classes under the name of zero-shot generalization. All studies revealed large performance degradations in the systematic generalization compared to the in-distribution generalization (generalization within the same combinations as the train data) unless some techniques are intentionally used. As explained in the previous subsection, HOI detection can be regarded as a subset of SGG focusing on measuring a model's capability for complex scene understanding. Thus, we regard the improvement of systematic generalization performance in HOI detection as an early step toward better models for SGG and other visual understanding tasks.

# 3 HICO-DET-SG and V-COCO-SG

In this section, we first explain the creation process of HICO-DET-SG and V-COCO-SG, the novel data splits for evaluating systematic generalization performance in HOI detection task. We then present the statistics of HICO-DET-SG and V-COCO-SG.

## 3.1 The creation process of the systematic generalization (SG) splits

When designing a pair of train data and test data of a systematic generalization (SG) split, we disallowed overlapping object–interaction combination classes between the train and test data so that HOI detection models are required to generalize to novel combinations. For example, in Figure 1, the train data consists of combinations such as <human, **wash**, car>, <human, **wash**, elephant>, <human, **walk**, horse>, and <human, **straddle**, horse>. An HOI detection model is tested whether it can generalize to novel combinations in the test data, such as <human, **wash**, horse>.

In the train data, we ensure that every object class is paired with multiple interaction classes, and that every interaction class is paired with multiple object classes. This split design makes it possible for a model to learn the concepts of object/interaction themselves independently from the specific interaction/object classes paired in the train data. To ensure that the test performance is not an artifact of a specific selection of combinations in the train and the test data, we prepared three distinct train-test splits of the HICO-DET-SG and V-COCO-SG, respectively.

When designing a test data, we eliminate images containing the same object-interaction combination classes as the train data. Consequently, the SG splits contain fewer images and HOIs in total than the original splits.

Table 1: Statistics of the HICO-DET-SG and V-COCO-SG as well as the original HICO-DET and V-COCO. The train and test data of the systematic generalization (SG) splits are composed of non-overlapping object-interaction combination classes. The test data of the SG splits contain fewer images and HOI triplets than the original test data because when designing the test data we eliminated images for which the object-interaction combination classes also exist in the train data.

| Data splits | # of images | | # of HOI triplets | | # of object-interaction comb. classes | |
| --- | --- | --- | --- | --- | --- | --- |
| | train | test | train | test | train | test |
| Original HICO-DET | 38,118 | 9,061 | 117,871 | 33,405 | 600 | 600 |
| HICO-DET-SG split1 | 38,312 | 8,515 | 119,331 | 14,475 | 540 | 60 |
| HICO-DET-SG split2 | 39,213 | 7,656 | 122,299 | 14,811 | 540 | 60 |
| HICO-DET-SG split3 | 40,672 | 6,229 | 120,096 | 8,994 | 540 | 60 |
| Original V-COCO | 5,400 | 4,946 | 14,153 | 12,649 | 228 | 228 |
| V-COCO-SG split1 | 7,297 | 2,850 | 18,214 | 3,872 | 160 | 68 |
| V-COCO-SG split2 | 7,057 | 3,066 | 15,644 | 4,322 | 160 | 68 |
| V-COCO-SG split3 | 6,210 | 3,888 | 10,951 | 6,244 | 160 | 68 |

The creation process of the SG splits is further detailed in Appendix A.

## 3.2 Statistics of the HICO-DET-SG and V-COCO-SG

The new HICO-DET-SG data splits were created based on the HICO-DET dataset (Chao et al., 2018) as explained above. The statistics of the original HICO-DET dataset and the HICO-DET-SG data splits are given in Table 1 (upper half). The original HICO-DET dataset contains 80 classes of objects, 117 classes of interactions, and 600 classes of object-interaction combinations. In the HICO-DET-SG splits, 540 out of 600 object-interaction combination classes are assigned to the train data; the remaining 60 classes are assigned to the test data.

The new V-COCO-SG data splits were created by the same process based on the V-COCO dataset (Gupta & Malik, 2015). The statistics of the original V-COCO dataset and the V-COCO-SG splits are given in Table 1 (lower half). The original V-COCO dataset contains 80 classes of objects, 29 classes of interactions, and 228 classes of object-interaction combinations. In the V-COCO-SG splits, 160 out of 228 object-interaction combination classes are assigned to the train data; the remaining 68 classes are assigned to the test data.

# 4 Experimental setups for evaluating representative HOI detection models

This section describes the experimental setups for evaluating the systematic generalization ability of recent representative HOI detection models. Subsection 4.1 explains the evaluated models and the reasons for their selection. Subsection 4.2 explains the experimental conditions in detail.

## 4.1 HOI detection models

We evaluated the systematic generalization performance of four representative HOI detection models: HOTR (Kim et al., 2021), QPIC (Tamura et al., 2021), FGAHOI (Ma et al., 2023), and STIP (Zhang et al., 2022). The characteristics of the four models are shown in Table 2. All models except STIP adopt the recently popularized one-stage architecture. The backbone (feature extraction) network of each model was pretrained on object detection task using the Microsoft COCO dataset (Lin et al., 2014). The encoders and decoders of HOTR, QPIC, and STIP were also pretrained on object detection task using the COCO dataset. However, the encoder and decoder of FGAHOI cannot be pretrained on object detection task because there are some modifications compared to its base object-detection model, namely, deformable DETR (Zhu et al., 2021). Therefore, we report the results of HOTR, QPIC, and STIP both with and without pretraining the encoder and decoder on the object detection task in Figure 2 of Section 5 and Tables 3 and 4 of Appendix B for the sake of a fair comparison. The details of each model are given below.

Table 2: Characteristics of the four HOI detection models: HOTR, QPIC, FGAHOI, and STIP. The upper half of the table describes the architecture types, feature extractors, and base models of the four models. The lower half gives the mean average precision (mAP, where higher values are desired) on the original HICO-DET and V-COCO reported in the original papers of each model.

|  | HOTR | QPIC | FGAHOI | STIP |
|---|---|---|---|---|
| Architecture type | One-stage parallel | One-stage end-to-end | One-stage end-to-end | Two-stage |
| Feature extractor | CNN | CNN | Multi-scale Transformer | CNN |
| Base model | DETR | DETR | deformable DETR | DETR |
| mAP (%) on original HICO-DET | 25.73 | 29.90 | 37.18 | 32.22 |
| mAP (%) on original HICO-DET (rare)* | 17.34 | 23.92 | 30.71 | 28.15 |
| mAP (%) on original V-COCO | 63.8 | 61.0 | 61.2 | 70.65 |

* The few-shot generalization performance evaluated on the rare-triplets (see Subsection 2.2).

**HOTR.** HOTR (Human-Object interaction detection TRansformer) (Kim et al., 2021) is among the first Transformer-based models for HOI detection. This model adopts a one-stage parallel architecture and consists of a CNN-based backbone (feature extractor), a shared encoder, an instance (human + object) decoder, and an interaction decoder. To get the matching between instance and interaction decoder outputs, three independent feed-forward networks, named HO Pointers, are trained to predict the correct <human, **interaction**, object> combination matching. Most part of the network (except HO Pointers) is based on DETR (DEtection TRansformer) (Carion et al., 2020), a Transformer-based object detector. Therefore, we can pretrain the backbone, shared encoder, instance decoder, and interaction decoder with DETR weights trained on the object detection task.

**QPIC.** QPIC (Query-based Pairwise human-object interaction detection with Image-wide Contextual information) (Tamura et al., 2021) is another Transformer-based HOI detection model proposed around the same time as HOTR. Like HOTR, it is based mainly on DETR but unlike HOTR, it adopts a one-stage end-to-end architecture with a single decoder and no HO Pointers. The backbone, encoder, and decoder can be pretrained using the weights of DETR trained on the object detection task.

**FGAHOI.** FGAHOI (Fine-Grained Anchors for Human-Object Interaction detection) (Ma et al., 2023) had exhibited the best performance on the HICO-DET dataset on the Papers with Code leaderboard [1] at the time of our experiments. The encoder of FGAHOI is trained to generate query-based anchors representing the points with high objectness scores in an image. To improve the computational efficiency, the decoder of FGAHOI predicts objects and interactions on the anchors alone and uses deformable DETR (Zhu et al., 2021), a modified version of DETR that computes self-attention from a limited range of feature maps. Therefore, the model can extract multi-scale feature maps from an image. Although the basic components described above are based on those of QAHOI (Chen & Yanai, 2021), FGAHOI can generate more fine-grained anchors than its predecessor due to the combination of three novel components: a multi-scale sampling mechanism, a hierarchical spatial-aware merging mechanism, and a task-aware merging mechanism. A novel training strategy (stage-wise training) is also designed to reduce the training pressure caused by overly complex tasks done by FGAHOI. The backbone of the network can be pretrained because it is based on Swin Transformer (Liu et al., 2021), but the encoder and the decoder of FGAHOI cannot be pretrained because there are some modifications on the deformable DETR (object detection model). For this reason, the performances of FGAHOI are reported only for the non-pretrained encorder and decorder cases in Figure 2 of Section 5 and Tables 3 and 4 of Appendix B.

**STIP.** STIP (Structure-aware Transformer over Interaction Proposals) (Zhang et al., 2022) had exhibited the best performance on the V-COCO dataset on the Papers with Code leaderboard [2] at the time of our experiments. This model has a two-stage architecture to perform HOI set prediction from non-parametric

---

[1] https://paperswithcode.com/sota/human-object-interaction-detection-on-hico
[2] https://paperswithcode.com/sota/human-object-interaction-detection-on-v-coco

interaction queries detected by an independent instance detector. Therefore, the model can explore inter- and intra-interaction structures during early training epochs by fixing the correspondence between the interaction query and each target HOI. The backbone and object detector can both be pretrained using the weights of DETR pretrained on an object detection task because the first stage of the network is the same as DETR.

## 4.2 Pretraining, hyperparameters, and other conditions

We used the official source code of the four models taken from the publicly-available repositories (URLs are listed in the References).

The backbone (feature extraction) networks of all models were pretrained on the object detection task using the Microsoft COCO dataset (Lin et al., 2014) under the settings reported in the respective original papers. The feasibility of pretraining the encoder and decoder parts depends on the model structure. The FGAHOI results were obtained without pretraining the encoder and decoder, because (as explained previously), the encoder and decoder of FGAHOI cannot be pretrained. For a fair comparison, we report the results of HOTR, QPIC, and STIP both with and without pretraining the encoder and decoder on the object detection task using the COCO dataset, although the original papers encouraged the use of pretrained weights to optimize the performance of the original HOI detection task.

Adopting the hyperparameters reported in the original papers and the official code repositories, we trained each model as described below.

For HOTR, ResNet-50 (He et al., 2016) was used as the backbone, the number of both encoder and decoder layers was set to 6, the number of attention heads was set to 8, the number of query embeddings was set to 300, the hidden dimension of embeddings in the Transformer was set to 256, and the dropout rate was set to 0.1. The model was trained for 100 epochs using the AdamW optimizer (Loshchilov & Hutter, 2019) with a batch size of 2, an initial learning rate of $10^{-5}$ for the backbone network and $10^{-4}$ for the other networks, and a weight decay of $10^{-4}$. Both learning rates decayed after 80 epochs.

For QPIC, ResNet-101 (He et al., 2016) was used as the backbone, the number of both encoder and decoder layers was set to 6, the number of attention heads was set to 8, the number of query embeddings was set to 100, the hidden dimension of embeddings in the Transformer was set to 256, and the dropout rate was set to 0.1. The model was trained for 150 epochs using the AdamW optimizer with a batch size of 16, an initial learning rate of $10^{-5}$ for the backbone network and $10^{-4}$ for the other networks, and a weight decay of $10^{-4}$. Both learning rates decayed after 100 epochs. The hyperparameters of the Hungarian costs and loss weights related to the bounding box were 2.5 times larger than those unrelated to the bounding box.

For FGAHOI, Swin-Large$_+^*$ (Liu et al., 2021) was used as the backbone, the number of both encoder and decoder layers were set to 6, the number of attention heads was set to 8, the number of query embeddings was set to 300, the hidden dimension of embeddings in the Transformer was set to 256, and the dropout was not applied. The model was trained with the AdamW optimizer with a batch size of 16, an initial learning rate of $10^{-5}$ for the backbone network and $10^{-4}$ for the other networks, and a weight decay of $10^{-4}$. On HICO-DET and HICO-DET-SG, the base network was trained for 150 epochs and the learning rate was dropped from the 120th epoch during the first stage of training. Subsequent training was performed over 40 epochs with a learning rate drop at the 15th epoch. On V-COCO and V-COCO-SG, the base network was trained for 90 epochs and the learning rate was dropped from the 60th epoch during the first stage of training. Subsequent training was performed over 30 epochs with a learning rate drop at the 10th epoch.

For STIP, ResNet-50 (He et al., 2016) was used as the backbone, the number of both encoder and decoder layers were set to 6, the number of attention heads was set to 8, the number of query embeddings for object detector was set to 100, the number of queries for interaction decoder was set to 32, the hidden dimension of embeddings in the Transformer was set to 256, and the dropout rate was set to 0.1. The model was trained for 30 epochs using the AdamW optimizer with a batch size of 8 and a learning rate of $5 \times 10^{-5}$.

We trained and tested the following seven types of models: HOTR, QPIC, FGAHOI, and STIP without pretraining the encoder and decoder, and HOTR, QPIC, and STIP with pretraining the encoder and decoder. Training and testing were performed once on each split. One training required approximately 1-2 days using 4 NVIDIA V100 GPUs.

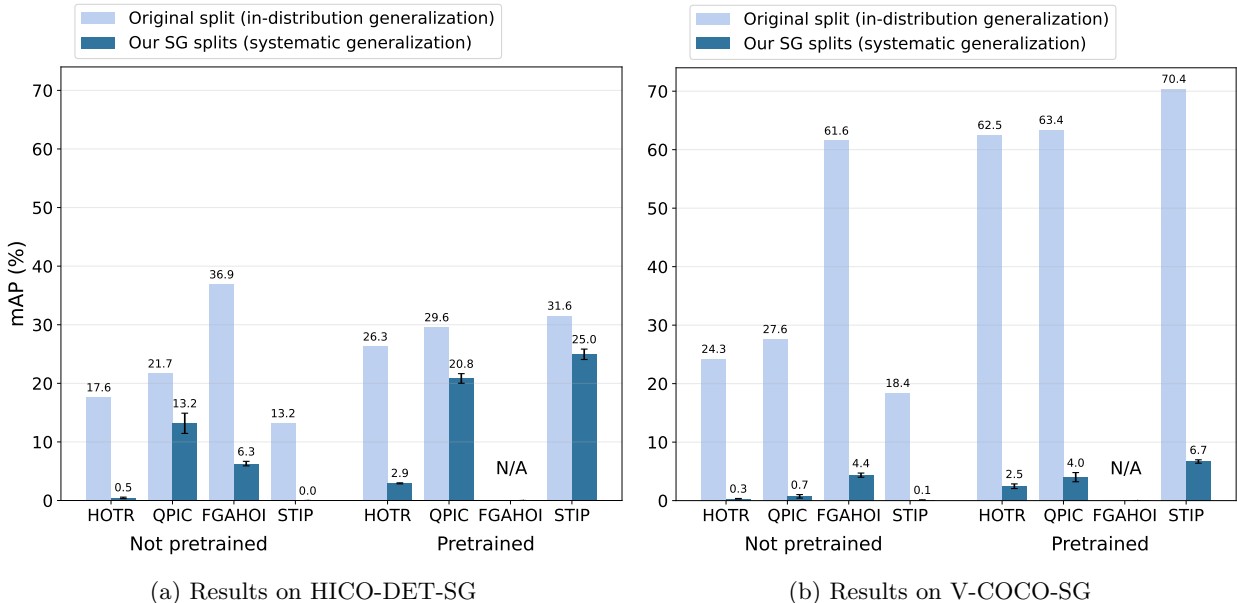

(a) Results on HICO-DET-SG
(b) Results on V-COCO-SG

Figure 2: Evaluation results of the systematic generalization performances on (a) HICO-DET-SG and (b) V-COCO-SG data splits, and the in-distribution generalization performances on the original splits. The mAPs (%) for the test data, which are the higher the better, of all models are considerably lower on both HICO-DET-SG and V-COCO-SG (dark blue) than on the original splits (pale blue) that evaluate the in-distribution generalization ability. The dark blue bars and the error bars represent the averages and the standard deviations, respectively, across the three distinct SG splits. "Not pretrained" denotes that the encoders and decoders of the models were trained from scratch, wherreas "Pretrained" denotes that the initial encoder and decoder weights were copied from DETR trained on object detection task using the Microsoft COCO dataset. The results of FGAHOI are reported only for "Not pretrained" cases because the encoder and decoder of FGAHOI cannot be pretrained on object detection task as explained in Section 4. Further details of the evaluation results are given in Tables 3 and 4 in Appendix B.

## 5 Evaluation results

This section reports the systematic generalization performances of the four representative HOI detection models (HOTR (Kim et al., 2021), QPIC (Tamura et al., 2021), FGAHOI (Ma et al., 2023), and STIP (Zhang et al., 2022)) evaluated on the HICO-DET-SG and V-COCO-SG. A qualitative inspection of failure cases is also presented.

### 5.1 Degradation in the systematic generalization performance

Figure 2 compares the evaluation results on HICO-DET-SG and V-COCO-SG with those on the original splits. The evaluation metric is the mean average precision (mAP), which was adopted in the original papers of the four models. A high mAP indicates that a model is well-performing. Regarding the HICO-DET-SG and V-COCO-SG, the averages and standard deviations calculated over the three distinct splits are presented in each subfigures.

The mAPs of all models are considerably lower on both HICO-DET-SG and V-COCO-SG (dark blue) than on the original splits (pale blue) that evaluate the in-distribution generalization ability. The differences among the test mAPs on the three SG splits are less than 3 percentage points, regardless of the model and base dataset. This means that the performances of all models largely degraded for any selection of object-interaction combinations in the train and test data. These results highlights the difficulty of systematic generalization in HOI detection task, i.e., recognizing novel combinations of known objects and interactions.

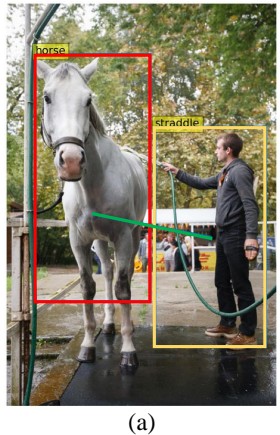 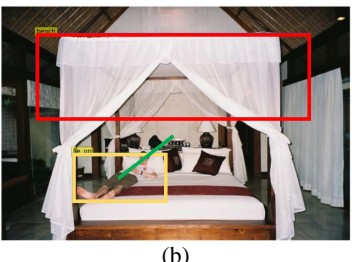 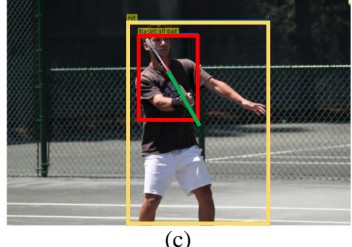

(a) (b) (c)

Figure 3: Three failure cases of STIP with the pretrained encoder and decoder after training and testing on HICO-DET-SG split3. (a) An example of predicting the wrong interaction class. The model predicted the interaction as **straddle**, although the correct class is **wash**. (b) An example of detecting the wrong object. The model predicted an irrelevant region as a wrong class bench, although it should detect a bed under the person. (c) An example of wrong class prediction of both object and interaction. The model predicted <human, **hit**, baseball bat> triplet although the correct answer is <human, **swing**, tennis racket>.

Further details of the evaluation results are given in Appendix B.

## 5.2 Qualitative inspection of failure cases

To further reveal the difficulty of systematic generalization in HOI detection, we inspected the failure cases. Figure 3 show the outputs of STIP with pretrained encoder and decoder trained and tested on HICO-DET-SG split3, which achieved the highest mAP among all models on all SG splits.

Figure 3 (a) shows an example of predicting the wrong interaction class, the most frequently observed error type. In this example, the model predicts the interaction as **straddle**, although the correct class is **wash**. The <human, **straddle**, horse> triplet appears in the train data but the <human, **wash**, horse> triplet appears only in the test data (the **wash** interaction appears with other objects in the train data). The model appears to predict the interaction from the object class (horse) alone and cannot generalize to the novel combination of <human, **wash**, horse>.

Figure 3 (b) shows an example of detecting the wrong object. The model is supposed to detect a bed under the person in the image but instead predicts an irrelevant region of the wrong class, bench. The <human, **lie on**, bench> triplet appears in the train data but the <human, **lie on**, bed> triplet appears only in the test data (bed appears with other interactions in the train data). Besides being unable to generalize to the novel combination, the model appears to predict the interaction (through its interaction decoder) mainly from visual cues of the human posture rather than from visual cues of the object or the positional relationships.

Figure 3 (c) shows an example of wrong class prediction of both object and interaction. The model predicted the tennis racket as a baseball bat and **swing** as **hit**. The <human, **hit**, baseball bat> triplet appears in the train data but the <human, **swing**, tennis racket> triplet appears only in the test data; moreover, the train data include the <human, **swing**, baseball bat> triplet but not the <human, **hit**, tennis racket> triplet. The model detected the object as a baseball bat at the first stage. Based on the detection result, the interaction was predicted as **hit** at the second stage, most-likely because the baseball bat frequently appeared with the **hit** interaction in the train data.

# 6 Discussion

## 6.1 Comparison of the results on HICO-DET-SG and V-COCO-SG

Comparing the results on the HICO-DET-SG and V-COCO-SG, we find that the performance gap is larger between HICO-DET (the original split) and HICO-DET-SG (the systematic generalization split) than between V-COCO and V-COCO-SG. This difference might reflect differences in the number of images and HOIs between the datasets. The HICO-DET-SG train data contains approximately 5.6 times as many images and 12.6 times as many HOIs as the train data of V-COCO-SG (Table 1). In more detail, the variety of object-interaction combination classes is 3.4 times higher in the HICO-DET-SG train data than in the V-COCO-SG train data, and more examples for one object-interaction combination exists in the HICO-DET-SG train data. In other computer vision tasks, increasing the diversity of train data is known to improve the systematic generalization performance (Lake & Baroni, 2018; Bahdanau et al., 2019; D'Amario et al., 2021; Madan et al., 2022). The same trend might be expected in HOI detection.

## 6.2 Comparison across models

The models achieved different performances on the SG splits. Without encoder and decoder pretraining, HOTR completely failed to generalize, as evidenced by the nearly 0% mAP. Even with the encoder and decoder pretrained on object detection task, the mAP of HOTR only improved to less than 3%. FGAHOI also underperformed on SG splits, with mAPs of approximately 6% or less. In contrast, QPIC showed some generalizability to novel combinations especially when using pretrained DETR weights: they achieved approximately 20% mAPs on the HICO-DET-SG splits. STIP with pretraining outperformed all other models on both the HICO-DET-SG and V-COCO-SG data splits. This superior performance might be attributed to the two-stage architecture of STIP, in which the instance and interaction detectors are independent and less affected by each other than in one-stage architectures. Supporting this inference, modular structures improved systematic generalization ability of deep neural networks in other computer vision tasks (Purushwalkam et al., 2019; Bahdanau et al., 2019; 2020; D'Amario et al., 2021; Madan et al., 2022; Yamada et al., 2023; Kamata et al., 2023).

## 6.3 Importance of pretraining the encoder and decoder

Pretraining the encoder and decoder on object detection task using the Microsoft COCO dataset improved the systematic generalization performances of HOTR, QPIC, and STIP. With pretraining, the mAP on HICO-DET improved from 0.4% to 2.7% for HOTR, from 12.1% to 20.8% for QPIC, and from 0.0% to 23.2% for STIP. Also the mAP on V-COCO changed from around 0.2% to around 2.1% for HOTR, from around 0.6% to around 3.8% for QPIC, and from around 0.0% to 6.3% for STIP. Note that the pretraining using the COCO dataset improved the systematic generalization performance not only on V-COCO-SG (that is based on COCO dataset) but also on HICO-DET-SG.

In general, vision Transformers require a large amount of training to achieve high performance when trained from scratch (Khan et al., 2021). Therefore, without pretraining, it is natural that the Transformer-based HOI detection models perform poorly on in-distribution generalization and even more poorly on systematic generalization. The performance of STIP was particularly degraded without pretraining, possibly because the number of training epochs (30) was much smaller for STIP than for the other models (100). FGAHOI could not be evaluated with its encoder and decoder pretrained, because the network is designed only for HOI detection task and it is hard to obtain pretrained weights using other tasks such as object detection. If we modify the final part of FGAHOI to enable training on other tasks, the systematic generalization performance of FGAHOI would presumably improve with the pretrained weights.

## 6.4 Toward the improvement of systematic generalization performance in HOI detection

Based on the above experimental results, we propose the following suggestions for improving the systematic generalization performance of future HOI detection models.

First, increasing the diversity of the train data (variety of the object-interaction combination classes) likely improve systematic generalization ability of HOI detection models. This possible solution is supported by the differences between HICO-DET-SG and V-COCO-SG and the results in other computer vision tasks (Lake & Baroni, 2018; Bahdanau et al., 2019; D'Amario et al., 2021; Madan et al., 2022) (see Subsection 6.1).

Second, introducing two-stage or other modular structures might improve the systematic generalization performance of HOI detection models. This suggestion is supported by the higher performance of the pretrained STIP than of the other models and by the results in other computer vision tasks (Purushwalkam et al., 2019; Bahdanau et al., 2019; 2020; D'Amario et al., 2021; Madan et al., 2022; Yamada et al., 2023; Kamata et al., 2023) (see Subsection 6.2). Two-stage model has also shown its effectiveness for systematic generalization in Scene Graph Generation (Lu et al., 2016), a task closely related to HOI detection (see Section 2), .

Third, our results also confirmed that pretraining the encoder and decoder improves the systematic generalization performance of HOI detection models (see Subsection 6.3). In this study, the encoder and decoder were initialized with the weights of DETR (Carion et al., 2020) trained on object detection task using the Microsoft COCO dataset. Pretraining on other related tasks such as Scene Graph Generation might also improve the systematic generalization performance of HOI detection models.

In addition, previous work (Lu et al., 2016; Kan et al., 2021) have shown that integrating commonsense knowledge from external natural language resources is effective for improving the systematic generalization performance in Scene Graph Generation. This is the fourth direction we think worth pursuing in HOI detection as well.

## 7 Conclusion

We created new splits of two HOI detection datasets, HICO-DET-SG and V-COCO-SG, whose train and test data consist of separate combinations of object-interaction classes for evaluating the systematic generalization performance of HOI detection models. The test performances of representative HOI detection models were considerably worse on our SG splits than on the original splits, indicating that systematic generalization is a challenging goal in HOI detection. We also analyzed the results and presented four possible research directions for improving the systematic generalization performance. We hope that our new data splits and presented analysis will encourage further research on systematic generalization in HOI detection.

**Reproducibility statement**

The JSON files determining HICO-DET-SG and V-COCO-SG and the source code that created the files are publicly available at a GitHub repository. For the review, they can be assessed via the following anonymized repository: `https://anonymous.4open.science/r/hoi_sg-58CE/`. The URLs of other existing assets (datasets and source code) used in this study are provided in the References. The experimental setups for performance evaluation of representative HOI detection models are described in Section 4.

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

# A  Further details of the creation process of systematic generalization (SG) splits

This Appendix explains the process of splitting the HICO-DET and V-COCO datasets for evaluating systematic generalization performance.

First, we decided the numbers of object-interaction combination classes in the train and test data. Initially we attempted to match their ratio to the ratio of HOI triplets in the original train and test data, but eventually included more combination classes in the train data (540 in HICO-DET-SG and 160 in V-COCO-SG) to ensure that every object class is paired with multiple interaction classes and that every interaction class is paired with multiple object classes. This makes it possible for a model to learn the concepts of object/interaction themselves independently of the specific interaction/object paired in the train data.

We then created the SG splits as described in Algorithm 1. From the test data, we eliminated images containing the same object–interaction combination classes as the train data. Consequently, the SG splits contain fewer images and HOI triplets in total than the original datasets (Table 1).

---

**Algorithm 1** Creation of the systematic generalization (SG) splits.

---

seed = 0
**while** flag **do**
    test_combinations = SELECT(all_combinations, seed)
    train_data = []
    test_data = []
    **for** scene in dataset **do**
        sum = 0
        test_hois = []
        **for** hoi in scene.hois **do**
            match = COUNT(test_combinations, [hoi.object_class, hoi.interaction_class])
            sum = sum + match
            **if** match > 0 **then**
                test_hois.append(hoi)
            **end if**
        **end for**
        **if** sum == length(scene.hois) **then**
            test_data.append(scene)
        **else if** sum == 0 **then**
            train_data.append(scene)
        **end if**
    **end for**
    **if** VERIFY(train_data) **then**
        flag = False
    **else**
        seed += 1
    **end if**
**end while**

---

Remark: "test_combinations" is a list of object-interaction combination classes to be contained only in the test data, "all_combinations" is a list of all combinations of object-interaction classes in the original dataset, "SELECT" is the function that randomly selects "test_combinations" from "all_combinations" with a specified seed, "dataset" represents the set of images in the original dataset (whole the train and test data) and their annotations, and "scene" represents a set of one image and HOI triplets in the image. The "COUNT" function returns the number of components in the first argument which is equal to the second argument. The "VERIFY" function verifies that all object and interaction classes alone are contained in the train data, that every object class is paired with multiple interaction classes, and that every interaction class is paired with multiple object classes. When creating another distinct split, the initial seed is set larger than the seed used for creating the existing split.

Finally, we verified that all object and interaction classes alone are contained in the train data, while certain combinations are contained only in the test data.

To ensure that the test performance is not an artifact of a specific selection of object-interaction combination classes in the train and test data, we prepared three distinct train-test splits for each of the HICO-DET-SG and V-COCO-SG.

The actual source code that created the SG splits is publicly available at the following anonymized repository for review: `https://anonymous.4open.science/r/hoi_sg-58CE/`.

## B  Further details of the evaluation results

To ensure that the models are well-trained, we evaluated the performances of the models on the train data before the evaluation on the test data. The left halves of Table 3 and Table 4 display the mAPs on the train data of HICO-DET-SG and V-COCO-SG, respectively. For both datasets and for all the models, the mAPs on the train data of the SG splits are coequal to or slightly higher than the original splits. This is probably attributed to the lower variety of triplets in the train data of the SG splits: HICO-DET-SG and V-COCO-SG contain 540 and 160 object-interaction combinations, respectively, out of 600 and 228 combinations contained in the original datasets, respectively. The right halves of these tables represent the raw data used for constructing Figure 2.

Table 3: Further details of the systematic generalization performances evaluated on HICO-DET-SG data splits. The mAPs (%) for both train and test data are given in this table. The right half of the table represents the raw data used for constructing Figure 2 (a). The values in brackets are those reported in the original papers. The term "pretraining" means that the initial weights of the model's encoder and decoder were copied from DETR (Carion et al., 2020) trained on object detection using the Microsoft COCO dataset (Lin et al., 2014). The results of FGAHOI are reported only for the non-pretrained cases due to the reasons given in Section 4. The test performances of all models are considerably lower on the HICO-DET-SG compared to original HICO-DET.

| | Evaluation on train data (reference) | | | | | | | | Evaluation on test data (main) | | | | | | | |
| | HOTR | | QPIC | | FGAHOI | STIP | | | HOTR | | QPIC | | FGAHOI | STIP | | |
| pretraining | x | ✓ | x | ✓ | x | x | ✓ | | x | ✓ | x | ✓ | x | x | ✓ | |
|---|---|---|---|---|---|---|---|---|---|---|---|---|---|---|---|---|
| Original HICO-DET (mAP in literature) | 30.54 | 44.80 | 33.29 | 46.28 | 55.82 | 13.47 | 32.23 | | 17.63 | 26.30 (25.73) | 21.70 | 29.59 (29.90) | 36.91 (37.18) | 13.21 | 31.57 (32.22) | |
| SG split1 | 33.92 | 46.03 | 34.05 | 49.70 | 58.97 | 13.69 | 33.17 | | 0.33 | 2.81 | 11.23 | 21.94 | 6.14 | 0.00 | 24.53 | |
| SG split2 | 31.48 | 42.04 | 30.23 | 48.28 | 55.95 | 15.13 | 34.44 | | 0.40 | 2.97 | 12.87 | 19.98 | 5.93 | 0.00 | 24.14 | |
| SG split3 | 32.05 | 44.91 | 35.54 | 47.90 | 54.72 | 13.25 | 30.61 | | 0.62 | 3.01 | 15.43 | 20.58 | 6.83 | 0.00 | 26.19 | |
| Average | 32.48 | 44.33 | 33.30 | 48.63 | 56.55 | 14.02 | 32.74 | | 0.45 | 2.93 | 13.18 | 20.83 | 6.30 | 0.00 | 24.95 | |

Table 4: Further details of the systematic generalization performances evaluated on V-COCO-SG data splits. The mAPs (%) for both train and test data are given in this table. The right half of the table represents the raw data used for constructing Figure 2 (b). The values in brackets are those reported in the original papers. The term "pretraining" means that the initial weights of the model's encoder and decoder were copied from DETR (Carion et al., 2020) trained on object detection using the Microsoft COCO dataset (Lin et al., 2014). The results of FGAHOI are reported only for the non-pretrained cases due to the reasons given in Section 4. The test performances of all models are considerably lower on the V-COCO-SG compared to original V-COCO for all the models.

| | Evaluation on train data (reference) | | | | | | | | Evaluation on test data (main) | | | | | | | |
| | HOTR | | QPIC | | FGAHOI | STIP | | | HOTR | | QPIC | | FGAHOI | STIP | | |
| pretraining | x | ✓ | x | ✓ | x | x | ✓ | | x | ✓ | x | ✓ | x | x | ✓ | |
|---|---|---|---|---|---|---|---|---|---|---|---|---|---|---|---|---|
| Original V-COCO (mAP in literature) | 28.23 | 64.72 | 30.61 | 65.63 | 64.27 | 19.10 | 72.89 | | 24.26 | 62.54 (63.8) | 27.64 | 63.41 (61.0) | 61.57 (61.2) | 18.43 | 70.43 (70.7) | |
| V-COCO-SG split1 | 30.57 | 65.79 | 31.24 | 67.25 | 67.49 | 23.51 | 71.91 | | 0.31 | 2.23 | 1.12 | 4.22 | 4.23 | 0.17 | 6.87 | |
| V-COCO-SG split2 | 31.53 | 67.28 | 32.53 | 68.43 | 69.28 | 20.04 | 74.38 | | 0.29 | 3.02 | 0.68 | 4.85 | 4.84 | 0.00 | 6.27 | |
| V-COCO-SG split3 | 28.21 | 61.07 | 29.83 | 60.47 | 63.24 | 22.41 | 73.43 | | 0.38 | 2.13 | 0.39 | 2.96 | 3.99 | 0.00 | 6.91 | |
| Average over SG splits | 30.10 | 64.71 | 31.20 | 65.38 | 66.67 | 21.99 | 73.24 | | 0.33 | 2.46 | 0.73 | 4.08 | 4.35 | 0.06 | 6.68 | |

