# OpenReview forum: "HICO-DET-SG and V-COCO-SG: New Data Splits for Evaluating the Systematic Generalization Performance of Human-Object Interaction Detection Models"
_TMLR — Rejected by TMLR_

### Review · Reviewer_8ZT7 · 2023-09-16

**Summary Of Contributions:**

This paper presents new data splits for HOI detection datasets HICO-DET and V-COCO. The key idea is to avoid training and testing on the same object-interaction pairs, which previous splits allowed. The new splits more convincingly evaluate the generalization capability of HOI models. The results show that model accuracies in the new splits are much lower, which is surprising and informative.

**Audience:**

Yes

**Claims And Evidence:**

Yes

**Requested Changes:**

"no open benchmarks or evaluation studies have been published for the systematic generalization in HOI detection"

This might be true, but are the authors aware of Bongard-HOI (CVPR 2022)? It's a different problem setup, but it does aim to measure generalization, by separating object-interaction pairs between train and test, as done here.

Typo:
HOI detection models are required systematic generalization -> HOI detection models require systematic generalization

**Strengths And Weaknesses:**

This paper is well written, the work is well motivated, and the results are interesting. Although I am not working in this area, I believe the work is useful to the community.

I appreciate the section summarizing potential ways to improve results, although none of these are novel (and are not claimed to be so).

A downside to the paper is that the contributions are quite narrow. Simplicity is good, but narrowness may mean limited interest. Still, for the people working on HOI (and particularly working on HICO-DET and V-COCO), I think the interest will be high.

---

> ### Author Response · Authors · 2023-11-16
> **Author rebuttal**
>
> Regarding Requested changes:
> Thank you for introducing the Bongard-HOI benchmark, we checked the paper. Bongard-HOI is designed to evaluate few-shot generalization performance for objects or actions which are unseen during training. Therefore, the aim is different from our SG splits, which evaluates systematic generalization performance (zero-shot generalization performance for unseen combinations of seen objects and actions). However, we are committed to enhancing the related work section by providing additional context and discussion on the Bongard-HOI benchmark.
>
> We also thank the reviewer for bringing a grammatical error to our attention.

---

### Review · Reviewer_Q6oA · 2023-10-26

**Summary Of Contributions:**

This paper proposes open benchmarks to investigate the systematic generalization of human-object interaction (HOI), termed HICO-DET-SG and V-COCO-SG, based on the HICO-DET and V-COCO datasets.
Meanwhile, the authors also evaluate some representative HOI methods on the new proposed benchmarks with and without additional pretrained data.
Last, the authors further introduce four possible directions on the HOI systematic generalization topic.

**Audience:**

Yes

**Broader Impact Concerns:**

This paper focuses on evaluating the HOI methods. The potential bias in the dataset may mislead the usage of the HOI methods.

**Claims And Evidence:**

No

**Requested Changes:**

Please see the above weaknesses part.

**Strengths And Weaknesses:**

Strengths:

**1**: The proposed benchmarks are practically useful since the generalization of objects and interaction classes is very important when applying the HOI methods.

**2**: The derived four possible directions for improving the systematic generalization performance of HOI methods could benefit the researchers.

Weaknesses:

**1**: Limited analysis of the proposed benchmarks. The authors only focus on the number of images and classes, as shown in Table 1. However, the distribution of interaction triplets is very important in this topic. For example, is there a long-tailed distribution? What's the distribution like for objects or the interaction? What's the distribution of relations between the objects and interactions? Since the proposed benchmark is regarded as the most important contribution in this paper, I think adding more in-depth analysis is necessary.

**2**: Limited evaluation. This paper misses some newly developed HOI methods based on vision-language models such as RLIPv1 [1] and RLIPv2 [2]. The language guidance can benefit from the large-scale pre-trained vision-language models and use more flexible text guidance to solve the HOI problem, especially in the systematic generalization topic that this paper mainly talks about. Therefore, I encourage the authors to add an analysis of the vision-language-based methods to the proposed benchmarks.

minor:

**1**: I find the results for comparing different methods in Table 2 and Figure 2 convey similar conclusions, i.e., existing methods perform well on existing benchmarks but are not good at the proposed benchmarks. So, I wonder the necessity of the existence of Table 2.

**2**: I find Section 4.2 (introducing the hyperparameters of existing methods) not helpful to help me understand the importance of the SG split dataset. I think it would be better if the experimental part focuses on evaluating the proposed benchmarks (different datasets and splits). The setup part can be moved to the appendix.


Overall, I suggest the authors reorganize the experimental part and convey more valuable insights into the systematic generalization problem of HOI.

[1] Yuan, Hangjie, et al. "Rlip: Relational language-image pre-training for human-object interaction detection." NeurIPS 2022.

[2] Yuan, Hangjie, et al., "RLIPv2: Fast Scaling of Relational Language-Image Pre-Training." ICCV 2023.

---

> ### Author Response · Authors · 2023-11-16
> **Author rebuttal**
>
> Regarding Weakness 1: We acknowledge the suggestion to include the distribution of objects, relations, and triplets. We will carefully consider integrating this information, while expecting minimal variation compared to the original dataset due to the random selection methodology for determining which triplets appear in train or test data.
>
> Regarding Weakness 2: Thank you for bringing our attention to these important literatures. We selected the models to be evaluated in our paper based on the fine-tuned best mAP. RLIPv1 was excluded from the evaluation because it exhibited lower performance in this criterion compared to QAHOI or FGAHOI. Regarding RLIPv2, it had not yet been published at the time of writing this paper. We checked the papers and found that they seem closely related to the fourth direction given in Section 6.4. We are afraid that it is difficult to conduct new experiments with these methods by Nov 30 (decision due), but we plan to mention them in Section 6.4. (We also think the claims written in Sections 5 and 6 are still valid without the new experiments with these methods.)
>
> Regarding minor comment 1: (We suppose that this comment is about Tables 3 and 4 instead of Table2.) While it is true that the contents presented on the right halves of the Table 3 and Table 4 (Evaluation on test data) are basically the same as those presented in the Figure 2, the inclusion of the left halves of the tables (Evaluation on train data) shows clear performance gaps between those evaluated on train and test data. This enables us to conclude that the lower performance on test data in the SG split than that in the original one is not due to insufficient training but overfitting to the object-interaction combinations in train data. In addition, Figure 2 shows only the mean and standard deviation of the accuracy obtained from the three distinct splits, while the Tables show raw data.
> We are considering adding notes in the captions of Tables 3 and 4 to clarify the purpose of presenting them.
>
> Regarding minor comment 2: Thank you for pointing this out. According to your advice, we plan to move the details of hyperparameter setting to Appendix.

---

### Review · Reviewer_x1Hr · 2023-11-02

**Summary Of Contributions:**

This paper presents several new data splits for the Human-Object Interaction (HOI) detection task, which aims to evaluate HOI models' ability to detect novel (unseen) object-interaction combinations. The paper shows that SOTA HOI models perform much worse on novel combinations than on the original test split. It conducts comprehensive comparisons and analysis, and provides some insights about possible future research directions.

**Audience:**

Yes

**Broader Impact Concerns:**

NA.

**Claims And Evidence:**

Yes

**Requested Changes:**

- A more comprehensive survey is suggested. More related work should be considered and included in this paper.
- Given the fact that existing HOI models perform much more poorly on test splits with novel combinations, it's better to introduce some inspiring methods to improve the zero-shot detection ability of novel object-interactions (with comparisons vs. existing HOI models).

**Strengths And Weaknesses:**

Strengths:

- This paper contributes new data splits for HICO-DET and V-COCO datasets, which can be used for evaluating the zero-shot (object-interaction combination) HOI detection ability.
- This paper is easy to follow and the compared HOI methods are well-illustrated. The comparative experiments are well-organized and the analysis is comprehensive.

Weaknesses:

- The novelty of this paper is limited and the survey of related works is insufficient. This paper is **definitely not the first one** to present new HOI data splits of novel (unseen) object-interaction combinations.
  - Hou et al. [1] proposed a visual compositional learning method that composes new interaction features (e.g., ride-horse) from known interaction concepts (e.g., feed-horse and ride-bicycle), and benefits the zero-shot HOI detection performance.
  - Recent work [2] provided a new split of HICO-DET and V-COCO datasets with novel object-interaction combinations. The authors in [2] even annotated extra possible object-interaction combinations. e.g., they extend the number of object-interaction concepts to  1,681 on HICO-DET dataset (compared to the original 600 concepts) , and 401 object-interaction concepts on V-COCO dataset (originally 228 concepts).
  - The SWiG-HOI [3] is a larger dataset than HICO-DET and V-COCO, which is not surveyed and discussed in this paper. It contains 1000 object classes and 406 action (interaction) classes, which can produce more object-interaction combinations than HICO-DET and V-COCO (consequently more novel/unseen combinations in its test split).

- The contribution of this paper is limited. First, creating new splits of HOI datasets is a relatively simple process. Moreover, this paper only compares several SOTA HOI methods in the proposed new splits of HOI datasets. It does not introduce any approach in terms of how to improve the zero-shot combination detection ability (or the systematic generalization ability as called in the paper).

- The insights about improving the systematic generalization (which is claimed as contributions) are somewhat trivial. E.g., increasing the diversity of the training data and pre-training the encoder & decoder on object detection tasks can improve the performance. It seems that these are known commonsense in the HOI detection community.

[1] Hou, Zhi, et al. "Visual compositional learning for human-object interaction detection." ECCV 2020.

[2] Hou, Zhi, Baosheng Yu, and Dacheng Tao. "Discovering human-object interaction concepts via self-compositional learning."  ECCV 2022.

[3] Wang, Suchen, et al. "Discovering human interactions with large-vocabulary objects via query and multi-scale detection." ICCV 2021.

---

> ### Author Response · Authors · 2023-11-16
> **Author rebuttal**
>
> Regarding Weaknesses 1.1 and 1.2: We express our gratitude for the introduction of the methods that tackle visual compositional learning to achieve better zero-shot HOI detection. Recognizing their relevance to our research, we will incorporate these methods into our related work section. One concern especially regarding the work of Hou et al. [2] is that we cannot find a detailed explanation in their paper or the actual code in their repository to reproduce their systematic generalization benchmark. This makes it difficult to ensure fair comparisons when evaluating systematic generalization performance among HOI detection methods. In contrast, our work establishes transparency by providing the HICO-DET-SG and V-COCO-SG splits, along with the code to generate them, for public access. It is intended to create a benchmark which is not only accessible, but also meaningful to the HOI detection community.
>
> Regarding Weakness 1.3: Thank you for introducing SWiG-HOI [3] dataset. Although it currently falls outside the scope of our paper, primarily due to the prevailing focus on HICO and V-COCO datasets in the HOI detection literature, we acknowledge the importance of this larger scale HOI dataset. We will refer this literature in the revised paper.
>
> Regarding Weakness 2 and Weakness 3: Regarding the limited scope of our contributions, we emphasize that the primary objective of our work is to provide a foundation for systematic generalization of HOI detection through the development of a meticulously designed data splits. While we recognize the significance of proposing novel methods, we would like to point out that Section 6 of our paper extensively analyzes elements contributing to performance. Specifically, in Section 6.4, we made four suggestions toward the improvement of systematic generalization performance in HOI detection. We consider these suggestions are valuable as they are supported both by comprehensive analysis of our own experiments and results regarding the systematic generalization in other computer vision tasks. Although not proposing a new method, we think the contents of this study can attract the interest of TMLR readers.
>
> Regarding Requested Change 1: Please refer to the response to Weaknesses 1.1, 1.2 and 1.3.
>
> Regarding Requested Change 2: Please refer to the response to Weaknesses 2 and 3.

---

### Decision · Action_Editor_HjPj · 2023-12-21

**Recommendation:** Reject

**Comment:**

This paper presents an effort to improve Human-Object Interaction (HOI) detection models by proposing new data splits, HICO-DET-SG and V-COCO-SG, for evaluating systematic generalization performance. However, the paper falls short in several key areas (please see the detailed breakdown as above). Firstly, the novelty of the approach is limited, as creating new data splits is a relatively straightforward task and does not constitute a significant advancement in the field. Secondly, the paper lacks a comprehensive validation of larger, more diverse benchmarks, which is crucial for establishing the effectiveness and generalizability of the proposed method. Additionally, the analysis presented is somewhat narrow, lacking in-depth comparisons and insights that could strengthen the paper's contributions.

Given these limitations, the paper does not meet the high standards of innovation and thoroughness expected by TMLR.

**Audience:**

The paper's focus on creating new data splits for evaluating systematic generalization in Human-Object Interaction (HOI) detection models could appeal to researchers and practitioners in the field.

**Claims And Evidence:**

Reviewers have raised concerns about the limited novelty of the paper and its insufficient validation on larger benchmarks. Here are detailed justifications:

1. Limited novelty: as raised by all three reviewers, data split for scene graphs or human-object interaction benchmark is definitely not a new topic. For example, reviewer x1Hr suggested many more related data split work. However, the authors only included them into the related work, but failed to provide more evaluations to justify the proposed split as a better design philosophy. Note that this limitation is confirmed by all the reviewers' final recommendation comments, I will quote them here:

x1Hr: The author's rebuttal did not fully address the original concerns on the limited novelty and insufficient validation on larger benchmarks. Given the overall limited contribution, I am leaning toward rejection.

Q6oA: The analysis/evaluation of the proposed HOI benchmark and comparisons with other HOI benchmarks (from other reviewers) are very limited. The replies and updates from the authors fail to provide any new results and explanations. Therefore, I remain my previous ideas and tend to reject this paper.

8ZT7: I am persuaded by reviewer x1Hr, pointing out that this is not the first work to present HOI data splits of novel object-interaction combinations. Even I was able to suggest a data split which is quite close (Bongard-HOI), despite not being an expert here. I also agree that "creating new splits of HOI datasets is a relatively simple process", which aligns with my point about the simplicity and narrowness of this paper. Overall I am not strongly opposed to this paper's acceptance, but given that none of the reviewers are particularly excited, it is hard to argue for it.

2. Limited benchmarks: due to the fast development of vision-language foundation models, the challenge of visual generalization (e.g., zero-shot recognition) may be fundamentally changed. However, the authors failed to provide sufficient evaluations for SOTA zero-shot methods such as various of prompt tuning methods.

Therefore, in summary, the paper's contributions are viewed as somewhat narrow, and some suggestions for creating new data splits for Human-Object Interaction (HOI) datasets, as presented in the paper, is a relatively simple process. Additionally, there are critiques about the paper's limited analysis and evaluation scope, suggesting that it could have included more comprehensive comparisons and insights.